# IMPROVING MODEL ROBUSTNESS WITH LATENT DISTRIBUTION LOCALLY AND GLOBALLY

## ABSTRACT

We propose a novel adversarial training method which leverages both the local and global information to defend adversarial attacks. Existing adversarial training methods usually generate adversarial perturbations locally in a supervised manner and fail to consider the data manifold information in a global way. Consequently, the resulting adversarial examples may corrupt the underlying data structure and are typically biased towards the decision boundary. In this work, we exploit both the local and global information of data manifold to generate adversarial examples in an unsupervised manner. Specifically, we design our novel framework via an adversarial game between a discriminator and a classifier: the discriminator is learned to differentiate the latent distributions of the natural data and the perturbed counterpart, while the classifier is trained to recognize accurately the perturbed examples as well as enforcing the invariance between the two latent distributions. We conduct a series of analysis on the model robustness and also verify the effectiveness of our proposed method empirically. Experimental results show that our method substantially outperforms the recent state-of-the-art (i.e. Feature Scattering) in defending adversarial attacks by a large accuracy margin (e.g. $17.0\%$ and $18.1\%$ on SVHN dataset, $9.3\%$ and $17.4\%$ on CIFAR-10 dataset, $6.0\%$ and $16.2\%$ on CIFAR-100 dataset for defending PGD20 and CW20 attacks respectively).

## 1 INTRODUCTION

Deep Neural Networks (DNNs) have achieved impressive performance on a broad range of datasets, yet can be easily fooled by adversarial examples or perturbations (LeCun et al., 2015; He et al., 2016; Gers et al., 1999). Adversarial examples have been shown to be ubiquitous beyond different tasks such as image classification (Goodfellow et al., 2014), segmentation (Fischer et al., 2017), and speech recognition (Carlini & Wagner, 2018). Overall, adversarial examples raise great concerns about the robustness of learning models, and have drawn enormous attention over recent years.

To defend adversarial examples, great efforts have been made to improve the model robustness (Kannan et al., 2018; You et al., 2019; Wang & Zhang, 2019; Zhang & Wang, 2019). Most of them are based on the adversarial training, i.e. training the model with adversarially-perturbed samples rather than clean data (Goodfellow et al., 2014; Madry et al., 2017; Lyu et al., 2015). In principle, adversarial training is a min-max game between the adversarial perturbations and classifier. Namely, the indistinguishable adversarial perturbations are designed to mislead the output of the classifier, while the classifier is trained to produce the accurate predictions for these perturbed input data. Currently, the adversarial perturbations are mainly computed by enforcing the output invariance in a supervised manner (Madry et al., 2017). Despite its effectiveness in some scenarios, it is observed recently that these approaches may still be limited in defending adversarial examples.

In particular, we argue that these current adversarial training approaches are typically conducted in a local and supervised way and fail to consider globally the overall data manifold information; such information however proves crucially important for attaining better generalization. As a result, the generated adversarial examples may corrupt the underlying data structure and would be typically biased towards the decision boundary. Therefore, the well-generalizing features inherent to the data distribution might be lost, which limits the performance of the DNNs to defend adversarial examples even if adversarial training is applied (Ilyas et al., 2019a; Schmidt et al., 2018). For illustration, we have shown a toy example in Figure 1. As clearly observed, adversarially-perturbed examples gen-

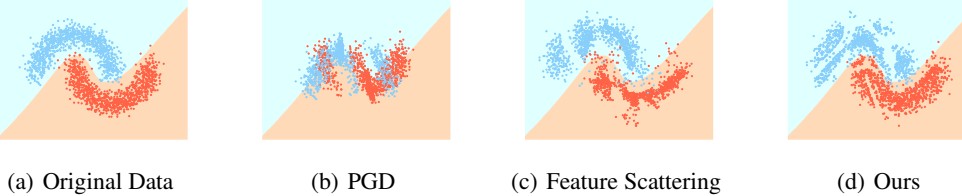

|     |     |     |     |
| --- | --- | --- | --- |
| (a) Original Data | (b) PGD | (c) Feature Scattering | (d) Ours |

Figure 1: Illustrative example of different perturbation schemes. (a) Original data, Perturbed data using (b) PGD: a supervised adversarial generation method (c) Feature Scattering, and (d) the proposed ATLD method. The overlaid boundary is from the model trained on clean data.

erated by PGD, one of the most successful adversarial training method, corrupt the data manifold, which would inevitably lead to poor performance if the training is conducted based on these perturbed examples. On the other hand, the current state-of-the-art method Feature Scattering (Zhang & Wang, 2019) can partially alleviate this problem but still leads to corruptions on the data manifold.

To address this limitation, we propose a novel method called Adversarial Training with Latent Distribution (ATLD) which additionally considers the data distribution globally in an unsupervised fashion. In this way, the data manifold could be well preserved, which is beneficial to attain better model generalization. Moreover, since the label information is not required when computing the adversarial perturbations, the resulting adversarial examples would not be biased towards the decision boundary. This can be clearly observed in Figure 1(d).

Our method can be divided into two steps: first, we train the deep model with the adversarial examples which maximize the variance between latent distributions of clean data and adversarial counterpart rather than maximizing the loss function. We reformulate it as *a minimax game between a discriminator and a classifier*. The adversarial examples are crafted by the discriminator to make different implicitly the latent distributions of clean and perturbed data, while the classifier is trained to decrease the discrepancy between these two latent distributions as well as promoting accurate classification on the adversarial examples as Figure 2 shows. Then, during the inference procedure, we generate the specific perturbations through the discriminator network to diminish the impact of the adversarial attack as shown in Figure 6 in Appendix.

On the empirical front, with the toy examples, we show that our proposed method can preserve more information of the original distribution and learn a better decision boundary than the existing adversarial training method. We also test our method on three different datasets: CIFAR-10, CIFAR-100 and SVHN with the famous PGD, CW and FGSM attacks. Our ATLD method outperforms the state-of-the-art methods by a large margin. e.g. ATLD improves over Feature Scattering (Zhang & Wang, 2019) by $17.0\%$ and $18.1\%$ on SVHN for PGD20 and CW20 attacks. Our method also shows a large superiority to the conventional adversarial training method (Madry et al., 2017), boosting the performance by $32.0\%$ and $30.7\%$ on SVHN for PGD20 and CW20 attacks.

## 2 RELATED WORK

**Adversarial Training.** Adversarial training is a family of techniques to improve the model robustness (Madry et al., 2017; Lyu et al., 2015). It trains the DNNs with adversarially-perturbed samples instead of clean data. Some approaches extend the conventional adversarial training by injecting the adversarial noise to hidden layers to boost the robustness of latent space (Ilyas et al., 2019b; You et al., 2019; Santurkar et al., 2019; Liu et al., 2019). All of these approaches *generate the adversarial examples by maximizing the loss function with the label information*. However, the structure of the data distribution is destroyed since the perturbed samples could be highly biased towards the non-optimal decision boundary (Zhang & Wang, 2019). Our proposed method has a similar training scheme with adversarial training by replacing clean data with the perturbed one. Nevertheless, our method generates the adversarial perturbations without the label information which weakens the impact of non-optimal decision boundary and can retain more information of the underlying data distribution.

**Manifold-based Adversarial Training.** Song et al. (2017) propose to generate the adversarial examples by projecting on a proper manifold. Zhang & Wang (2019) leverage the manifold information in the forms of inter-sample relationship within the batch to generate adversarial adversarial perturbations. Virtual Adversarial Training and Manifold Adversarial Training are proposed improve model generalization and robustness against adversarial examples by ensuring the local smoothness of the data distribution (Zhang et al., 2018; Miyato et al., 2017). Some methods are designed to enforce the local smoothness around the natural examples by penalizing the difference between the outputs of adversarial examples and clean counterparts (Kannan et al., 2018; Chan et al., 2020; Jakubovitz & Giryes, 2018). All of these methods just *leverage the local information* of the distribution or manifold. Differently, our method generates the perturbations additionally considering the structure of distribution globally.

**Unsupervised Domain Adversarial Training.** Domain Adversarial Training shares a training scheme similar to our method where the classifier and discriminator compete with each other (Odena et al., 2017; Long et al., 2018; Ganin et al., 2016). However, its objective is to reduce the gap between the source and target distributions in the latent space. The discriminator is used to measure the divergence between these two distributions in the latent space. The training scheme of our method is also based on competition between the classifier and discriminator. Different from the previous framework, the discriminator of our method is used to capture the information of distributions of adversarial examples and clean counterparts in the latent space which helps generate the adversarial perturbations.

**GAN-based Adversarial Training Methods.** Several GAN-based methods leverage GANs to learn the clean data distribution and purify the adversarial examples by projecting them on clean data manifold before classification (Meng & Chen, 2017; Metzen et al., 2017). The framework of GAN can also be used to generate the adversarial examples (Baluja & Fischer, 2018). The generator produces the adversarial examples to deceive both the discriminator and classifier; the discriminator and classifier attempt to differentiate the adversaries from clean data and produce the correct labels respectively. Some adversary detector networks are proposed to detect the adversarial examples which can be well aligned with our method (Gong et al., 2017; Grosse et al., 2017). In these works, a pretrained network is augmented with a binary detector network. The training of the pretrained network and detector involves generating adversarial examples to maximize their losses. Differently, our method generates the adversarial examples just to minimize the loss of the discriminator and feed them as the training set to the classifier. Such adversarial examples are deemed to induce most different latent representations from the clean counterpart.

## 3 BACKGROUND

### 3.1 ADVERSARIAL TRAINING

Let us first introduce the widely-adopted adversarial training method for defending against adversarial attacks. Specifically, it solves the following minimax optimization problem through training.

$$\min_{\theta}\{\mathbb{E}_{(x,y)\sim\mathcal{D}}[\max_{x'\in S_x} L(x',y;\theta)]\},\tag{1}$$

where $x \in \mathbb{R}^n$ and $y \in \mathbb{R}$ are respectively the clean data samples and the corresponding labels drawn from the dataset $\mathcal{D}$, and $L(\cdot)$ is the loss function of the DNN with the model parameter $\theta \in \mathbb{R}^m$. Furthermore, we denote the clean data distribution as $Q_0$, i.e. $x \sim Q_0$. , and denote $x' \in \mathbb{R}^n$ as perturbed samples in a feasible region $S_x \triangleq \{z : z \in B(x,\epsilon) \cap [-1.0, 1.0]^n\}$ with $B(z,\epsilon) \triangleq \{z : \|x - z\|_\infty \leq \epsilon\}$ being the $\ell_\infty$-ball at center $x$ with radius $\epsilon$. By defining $f_\theta(\cdot)$ as the mapping function from the input layer to the last latent layer, we can also rewrite the loss function of the DNN as $l(f_\theta(x), y)$ where $l(\cdot)$ denotes the loss function calculated from the last hidden layer of the DNN, e.g. the cross entropy loss as typically used in DNN.

Whilst the outer minimization can be conducted by training to find the optimal model parameters $\theta$, the inner maximization essentially generates the strongest adversarial attacks on a given set of model parameters $\theta$. In general, the solution to the minimax problem can be found by training a network minimizing the loss for worst-case adversarial examples, so as to attain adversarial robustness. Given a set of model parameters $\theta$, the commonly adopted solution to the inner maximization problem can lead to either one-step (e.g., FGSM) or multi-step (e.g., PGD) approach (Madry et al., 2017). In

particular, for a given single point $x$, the strongest adversarial example $x'$ at the $t$-th iteration can be iteratively obtained by the following updating rule:

$$x^{t+1} = \Pi_{S_x}(x^t + \alpha \cdot \text{sgn}(\nabla_x L(x^t, y; \theta))), \tag{2}$$

where $\Pi_{S_x}(\cdot)$ is a projection operator to project the inputs onto the region $S_x$, $\text{sgn}(\cdot)$ is the sign function, and $\alpha$ is the updating step size. For the initialization, $x^0$ can be generated by randomly sampling in $B(x, \epsilon)$.

It appears in (1) that each perturbed sample $x'$ is obtained individually by leveraging its loss function $L(x', y; \theta)$ with its label $y$. However, without considering the inter-relationship between samples, we may lose the global knowledge of the data manifold structure which proves highly useful for attaining better generalization. This issue has been studied in a recent work (Zhang & Wang, 2019) where a new method named feature scattering made a first step to consider the inter-sample relationship *within the batch*; unfortunately this approach did not take the full advantages of the global knowledge of the entire data distribution. In addition, relying on the maximization of the loss function, the adversarially-perturbed data samples may be highly biased towards the decision boundary, which potentially corrupts the structure of the original data distribution, especially when the decision boundary is non-optimal (see Figure 1 again for the illustration).

## 3.2 DIVERGENCE ESTIMATION

To measure the discrepancy of two distributions, statistical divergence measures (e.g., Kullback-Leibler and Jensen-Shannon divergence) have been proposed. In general, given two distributions $\mathbb{P}$ and $\mathbb{Q}$ with a continuous density function $p(x)$ and $q(x)$ respectively, $f$-divergence is defined as $D_f(\mathbb{P}||\mathbb{Q}) \triangleq \int_{\mathcal{X}} q(x) f\left(\frac{p(x)}{q(x)}\right) dx$. The exact computation of $f$-divergence is challenging, and the estimation from samples has attracted much interest. For instance, leveraging the variational methods, Nguyen et al. (2010) propose a method for estimating $f$-divergence from only samples; Nowozin et al. (2016) extend this method by estimating the divergence with learning the parameters of discriminator. Specifically, the $f$-divergence between two distributions $\mathbb{P}$ and $\mathbb{Q}$ can be lower-bounded using Fenchel conjugate and Jensen's inequality (Nowozin et al., 2016).

$$\begin{aligned} D_f(\mathbb{P}||\mathbb{Q}) &= \int_{\mathcal{X}} q(x) \sup_{t \in \text{dom}f^*} \{t\frac{p(x)}{q(x)} - f^*(t)\}dx \\ &\geq \sup_{T \in \tau}(\int_{\mathcal{X}} p(x)T(x)dx - \int_{\mathcal{X}} q(x)f^*(T(x))dx) \\ &= \sup_{W}(\mathbb{E}_{x \sim \mathbb{P}}[g_f(V_W(x))] + \mathbb{E}_{x \sim \mathbb{Q}}[-f^*(g_f(V_W(x)))]), \end{aligned} \tag{3}$$

where $V_W : \mathcal{X} \to \mathbb{R}$ is a discriminator network with parameter $W$ and $g_f : \mathbb{R} \to \text{dom}f^*$ is an output activation function which is determined by the type of discriminator. $\tau$ is an arbitrary class of functions $T : \mathcal{X} \to \mathbb{R}$. $f$ is a convex lower-semicontinous function and $f^*$ is its conjugate defined by $f^*(t) = \sup_{u \in \text{dom}f}[ut - f(u)]$. The objective of discriminator for GANs is a special case of (3) with the activation function $g_f(t) = -\log(1 + e^{-t})$ and $f^*(g) = -\log(2 - e^g)$. It approximates the Jense-Shannon divergence between real and fake distributions. Arjovsky et al. (2017) also develop a method to estimate the Wasserstein-distance by neural network. In this paper, these methods will be used to estimate the Jensen-Shannon divergence between latent distributions induced by adversarial and clean examples.

## 4 ADVERSARIAL TRAINING WITH LATENT DISTRIBUTION

As discussed in Section 3.1, the conventional adversarial training methods rely on the knowledge of data labels. As a result, the local information to generate adversarial examples may be biased toward the decision boundary; such individual adversarial example generation does not capture the global knowledge of the data manifold.

To alleviate these limitations, we propose a novel method to compute the perturbed samples by leveraging the global knowledge of the whole data distribution and then disentangling them from the data labels and the loss function. Generally speaking, the perturbations are generated to enlarge the variance between latent distributions induced by clean and adversarial data.

Formally, we try to identify the set of adversarial examples $X_{adv}$ that yield in the latent space a distribution $P_\theta^*$ through $f_\theta(\cdot)$ that is the most different from the latent distribution $Q_\theta$ induced by the clean samples $X_{org} = \{x : x \sim Q_0\}$, without resorting to the corresponding labels $Y$. In other words, *the resulting adversarial examples can be deemed as manifold adversarial examples, which 'deceive' the manifold rather than fool the classifier as defined in the traditional adversarial examples.* It is noted that the latent space to be perturbed could be any hidden layer though it is defined in the last hidden layer before the softmax of a DNN in this paper. The optimization problem of the proposed adversarial training can then be reformulated as follows:

$$\min_\theta \quad \{\mathbb{E}_{f_\theta(x^{adv}) \sim P_\theta^*}[l(f_\theta(x^{adv}), y)] + D_f(P_\theta^* || Q_\theta)\} \tag{4}$$

$$\text{s.t.} \quad P_\theta^* = \arg \max_{P_\theta \in \mathcal{P}} [D_f(P_\theta || Q_\theta)] \tag{5}$$

where $l(\cdot)$ and $y$ are similarly defined as before, and $D_f(\cdot)$ is the $f$-divergence measure of two distributions. $\mathcal{P} = \{P : f_\theta(x') \sim P \quad \text{subject to} \quad \forall x \sim Q_0, x' \in B(x, \epsilon)\}$ is the feasible region for the latent distribution $P_\theta$ which is induced by the set of perturbed examples $X_p$ through $f_\theta(\cdot)$. $f_\theta(x')$ and $f_\theta(x^{adv})$ represents the latent features of the perturbed example $x'$ and adversarial example $x^{adv}$ respectively. Intuitively, we try to obtain the worst latent distribution $P_\theta^*$ which is induced by $X_{adv}$ through $f_\theta(\cdot)$ within the region $\mathcal{P}$, while the model parameter $\theta$ is learned to minimize the classification loss on the latent feature $f_\theta(x^{adv}) \sim P_\theta^*$ (or equivalently adversarial example $x^{adv} \in X_{adv}$) and the $f$-divergence between the latent distributions $P_\theta^*$ and $Q_\theta$ induced by adversarial examples $X_{adv}$ and clean data $X_{org}$.

It is still challenging to solve the above optimization problem, since both the objective function and the constraint are entangled with the adversarial examples $X_{adv}$ and the model parameters $\theta$. To make the problem more tractable, we propose a novel Adversarial Training with Latent Distribution (ATLD) method. In the next subsection, by taking into account the entire data distribution globally, we first focus on the constraint and identify the adversarial samples $X_{adv}$ through the maximization problem. We then solve the minimization of the objective function with the adversarial training procedure. To further enhance the performance, we add specific perturbations named Inference with Manifold Transformation (IMT) in Section 4.2 to input samples for enforcing them towards the more separable natural data manifold. Finally, we classify the transformed data points with the adversarially-trained model.

## 4.1 GENERATING ADVERSARIAL EXAMPLES FOR TRAINING

First, we optimize the constraint (5) to generate the adversarial examples or its induced distribution $P_\theta^*$ for training. Intuitively, the adversarial examples $X_{adv}$ are crafted to maximize the divergence between the latent distributions induced by natural examples $X_{org}$ and adversarial counterpart $X_{adv}$ in an unsupervised fashion since no knowledge of labels $Y$ is required. Together with the objective function in (4), our proposed adversarial training method is to minimize such divergence as well as the classification error for adversarial examples $X_{adv}$.

However, it is a challenging task to evaluate the divergence between two latent distributions. To make it more tractable, we leverage *a discriminator* network for estimating the Jensen-Shannon divergence between two distributions $P_\theta^*/P_\theta$ and $Q_\theta$ according to Section 3.2. It is noted again that the class label information is not used for generating adversarial examples. Hence the adversarial examples are still generated in an unsupervised way. Then, by using (3), the optimization problem in (4) and (5) can be approximated as follows in a tractable way.

$$\min_\theta \left\{ \underbrace{\sum_{i=1}^N L(x_i^{adv}, y_i; \theta)}_{L_f} + \sup_W \sum_{i=1}^N \underbrace{[\log D_W(f_\theta(x_i^{adv})) + (1 - \log D_W(f_\theta(x_i)))]}_{L_d} \right\} \tag{6}$$

$$\text{s.t.} \quad x_i^{adv} = \arg \max_{x_i' \in B(x_i, \epsilon)} \underbrace{[\log D_W(f_\theta(x_i')) + (1 - \log D_W(f_\theta(x_i)))]}_{L_d}$$

where $N$ denotes the number of training samples and $D_W$ denotes the discriminator network with the sigmoid activation function and parameter $W$. $f_\theta(x_i)$ is the latent feature of the clean sample $x_i$. $D_W$ is used to determine whether the latent feature is from adversary manifold (output the

manifold label of the latent feature). For ease of description, we represent the components in Eq. (6) as two parts: $L_f$ and $L_d$. $L_d$ is the manifold loss and $L_f$ represents the loss function of the classifier network.

We now interpret the above optimization problem. By comparing Eq. (6) and Eq. (4), it is observed that the Jensen-Shannon divergence between $P_\theta^*$ and $Q_\theta$ is approximated by $\sup_W \sum_{i=1}^N L_d$, and the minimization of the classification loss on adversarial examples is given by $\min_\theta \sum_{i=1}^N L_f$. The problem (6) is optimized by updating parameters $\theta$ and $W$ and crafting adversarial examples $\{x_i^{adv}\}_{i=0}^N$ iteratively. The whole training procedure can be viewed as the game among three players: the classifier, discriminator, and adversarial examples. The discriminator $D_W$ is learned to differentiate the latent distributions of the perturbed examples and clean data via maximizing the loss $L_d$ while the classifier $f_\theta$ is trained to (1) enforce the invariance between these two distributions to confuse the discriminator $D_W$ by minimizing the loss $L_d$, and (2) classify the adversarial examples as accurately as possible by minimizing $L_f$. For each training iteration, the adversarial examples are crafted to make different the adversarial latent distribution and natural one by maximizing $L_d$. Although $D_W$ cannot measure the divergence between the two latent distributions exactly at the first several training steps, it can help evaluate the divergence between distributions induced by perturbed examples and clean ones when the parameters $W$ converges.

However, when the latent distributions are multi-modal, which is a real scenario due to the nature of multi-class classification, it is challenging for the discriminator to measure the divergence between such distributions. Several work reveals that there is a high risk of failure in using the discriminator networks to measure only a fraction of components underlying different distributions (Arjovsky & Bottou, 2017; Che et al., 2016). Ma (2018) also shows that two different distributions are not guaranteed to be identical even if the discriminator is fully confused. To alleviate such the problem, we additionally train the discriminator $D_W$ to predict the class labels for latent features as (Odena et al., 2017; Long et al., 2018). As a result, the problem (6) can then be reformulated as:

$$\min_\theta \left\{ \sum_{i=1}^N \underbrace{L(x_i^{adv}, y_i; \theta)}_{L_f} + \sup_W \sum_{i=1}^N \underbrace{[\log D_W^0(f_\theta(x_i^{adv})) + (1 - \log D_W^0(f_\theta(x_i)))]}_{L_d^0} \right.$$
$$\left. + \min_W \underbrace{[l(D_W^{1:C}(f_\theta(x_i)), y_i) + l(D_W^{1:C}(f_\theta(x_i^{adv})), y_i)]}_{L_d^{1:C}} \right\} \tag{7}$$
$$\text{s.t.} \quad x_i^{adv} = \arg \max_{x_i' \in B(x_i, \epsilon)} \underbrace{[\log D_W^0(f_\theta(x_i')) + (1 - \log D_W^0(f_\theta(x_i)))]}_{L_d^0}$$

Here $D_W^0$ is the first dimension of the output of the discriminator, which indicates the manifold label of the latent features; $D_W^{1:C}$ are the remaining $C$ dimensions of the output of $D_W$, used to output the class label of the latent feature; $C$ denotes the number of classes, and $L_d^0$ and $L_d^{1:C}$ are the manifold loss and the classification loss for the discriminator network respectively. (*The detailed derivation for Eq. (6) and Eq. (7) can be seen in Appendix.*) The detailed training procedure of our framework is depicted in Figure 2.

**Remarks.** It is worth noting that the labeled information is not required for generating adversarial examples. Therefore, our method prevents the perturbed examples from highly biasing towards the decision boundary and more information of the original distribution structure is preserved. In addition, since the discriminator is trained with the whole dataset (both clean and adversarial examples), it captures the global information of data manifold. Consequently, by training with adversarial examples generated according to the manifold loss of the discriminator, our method can improve the model robustness against adversarial examples with the global structure of data distribution.

## 4.2 INFERENCE WITH MANIFOLD TRANSFORMATION

To enhance the generalization of ATLD, we further develop a new inference method with manifold transformation. Although adversarially-trained models can well recognize adversarial examples, there are still potential examples which are easily misclassified especially for unseen data. In other words, the generalization to adversarial examples is hard to achieve due to the more complex distribution of adversarial examples (Schmidt et al., 2018; Zhai et al., 2019). To alleviate this problem,

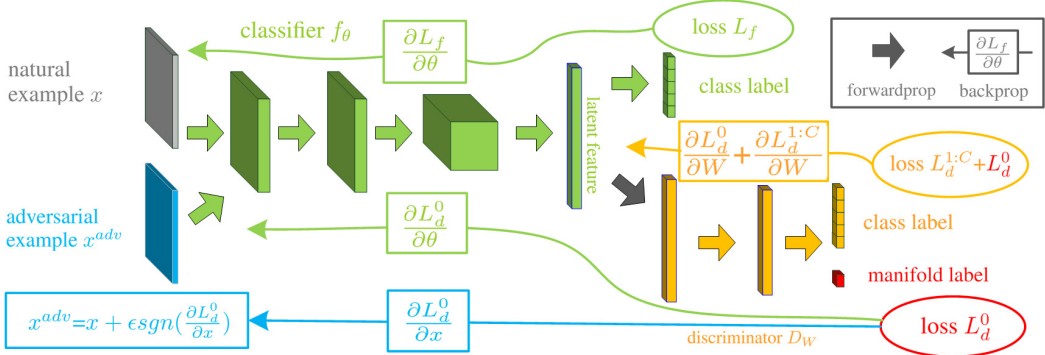

Figure 2: Overall architecture of ATLD and its training procedure. 1) The natural example is fed into the network, and the discriminator outputs its prediction. The manifold loss $L_d^0$ is computed with the prediction and true label and generates the adversarial example $x^{adv}$ (blue arrow). 2) Both the clean and adversarial sample are fed into the network to train the classifier (green arrow) and the discriminator (yellow arrow) iteratively.

our proposed inference method first pushes adversarial examples towards the manifold of natural examples which is simpler and further away from the decision boundary than the adversarial distribution. Then the more separable adjusted examples are classified by the adversarially-trained model. Specifically, the input sample is fed into our adversarially-trained model and the discriminator outputs the probability of such a sample lying on the adversarial manifold. If this probability is higher than a certain threshold, we compute the transformed example $x^t$ by adding the specific perturbation $r^*$ to the input sample $x$ to reduce such a probability. This perturbation can be computed as:

$$r^* = \arg \min_{\|r\|_\infty \leq \epsilon} \log D_W^0(f_\theta(x + r)). \tag{8}$$

Intuitively, the reduction of probability of this data point lying on adversarial manifold indicates that this point moves towards the benign example manifold after adding perturbation $r^*$. In other words, it becomes more separable since the benign example manifold is further away from the decision boundary. When the probability of the image lying on adversary manifold is lower than threshold, we still add such a perturbation to input image to make it more separable but with smaller magnitude. In the experiment part, we show this perturbation can move the adversarial examples away from the decision boundary. The whole inference procedure can be seen in Figure 5 in Appendix.

## 5 EXPERIMENTS

We conduct experiments on the widely-used datasets, e.g., CIFAR-10, SVHN, and CIFAR-100. Following the Feature Scattering method (Zhang & Wang, 2019), we leverage the wideresnet (Zagoruyko & Komodakis, 2016) as our basic classifier and discriminator model structure. During the training phase, the initial learning rate is empirically set to $0.1$ for all three datasets. We train our model 400 epochs with the transition epoch $60, 90$ and the decay rate $0.1$. The input perturbation budget is set to $\epsilon = 8$ with the label smoothing rate as $0.5$. We use $L_\infty$ perturbation in this paper including all the training and evaluation.

We evaluate the various models on white-box attacks and black-box attacks. Under the white-box attacks, we compare the accuracy of the proposed method with several competitive methods, including: (1) the original wideresnet (Standard) trained with natural examples; (2) Traditional Adversarial Training with PGD (AT) (Madry et al., 2017); (3) Triplet Loss Adversarial training (TLA) (Mao et al., 2019); (4) Layer-wise Adversarial Training (LAT): injecting adversarial perturbation into the latent space (Sinha et al., 2019); (5) Bilateral: adversarial perturb on examples and labels both (Wang & Zhang, 2019); (6) Feature-scattering: generating adversarial examples with considering inter-relationship of samples (Zhang & Wang, 2019). These comparison algorithms present the most competitive performance on defending adversarial attack. Under the black-box attacks, we compare four different algorithms used to generate the test time attacks: Vanilla training with natural examples, adversarial training with PGD, Feature Scattering, and our proposed model.

## 5.1 DEFENDING WHITE-BOX ATTACKS

We show the classification accuracy under several white-box attacks on CIFAR-10, CIFAR-100, SVHN in this section. We first report the accuracy on CIFAR-10 in Table 1 with the attack iterations $T = 20, 40, 100$ for PGD (Madry et al., 2017) and CW (Carlini & Wagner, 2017). We also conduct more experiments to further evaluate the robustness of our proposed method against more recent attacks, e.g. AutoAttack (Croce & Hein, 2020) and RayS (Chen & Gu, 2020)) as shown in Appendix B.2. As observed, overall, our proposed method achieves a clear superiority over all the defence approaches on both the clean data and adversarial examples (except that it is slightly inferior to Feature Scattering in FGSM). We also observe one exception that the standard model performs the best on clean data. Our approach performs much better than the other baseline models on PGD and CW attack. Particularly, we improve the performance of the recent state-of-the-art method Feature Scattering almost 3.1% and 5.2% under PGD20 and CW20 attack respectively. With the implementation of Inference with Manifold Transformation (IMT), our approach (ATLD-IMT) is 8.9% and 17.4% higher than the Feature Scattering under PGD20 and CW20 attack respectively. However, the performance on clean data is declined from 93.3% to 86.4% since IMT appears to have a negative effect for classifying clean data. In order to reduce the impact of IMT on the natural data, a threshold is used to limit the perturbation of IMT based on the output of discriminator. The perturbation is halved if the output of discriminator is within the range of $[0.3, 0.7]$ (ATLD-IMT+). Under such setting, our approach could achieve high performance on adversarial attacks without sacrificing its accuracy on clean data.

Similarly, the accuracy on CIFAR-100 and SVHN are shown in Table 2 with the attack iterations $T = 20, 100$ for both PGD and CW for conciseness. Although our method is slightly weaker than Feature Scattering under FGSM attack on CIFAR-100, overall, our proposed method ATLD achieves state-of-the-art performance over all the other approaches under various adversarial attacks. Furthermore, our ATLD-IMT version exceeds Feature Scattering by almost 19.2% and 23.8% against the attack of CW100 on CIFAR-100 and SVHN respectively. More details about the defense of white-box attacks under different attack budgets can be seen in Appendix.

Table 1: Accuracy under different White-box Attack attack on CIFAR-10

| MODELS | CLEAN | ACCURACY UNDER WHITE-BOX ATTACK ($\epsilon = 8$) | | | | | | |
|---|---|---|---|---|---|---|---|---|
| | | FGSM | PGD20 | PGD40 | PGD100 | CW20 | CW40 | CW100 |
| STANDARD | **95.60** | 36.90 | 0.00 | 0.00 | 0.00 | 0.00 | 0.00 | 0.00 |
| AT | 85.70 | 54.90 | 44.90 | 44.80 | 44.80 | 45.70 | 45.60 | 45.40 |
| TLA | 86.21 | 58.88 | 51.59 | - | - | - | - | - |
| LAT | 87.80 | - | 53.84 | - | 53.04 | - | - | - |
| BILATERAL | 91.20 | 70.70 | 57.50 | – | 55.20 | 56.20 | – | 53.80 |
| FS | 90.00 | 78.40 | 70.50 | 70.30 | 68.60 | 62.40 | 62.10 | 60.60 |
| ATLD | 93.34 | **87.91** | 73.58 | 72.95 | 72.82 | 67.63 | 66.26 | 65.40 |
| ATLD-IMT | 86.42 | 84.62 | 79.48 | **77.06** | 74.20 | **79.81** | 77.14 | **75.20** |
| ATLD-IMT+ | 90.78 | 84.37 | **79.82** | 76.71 | **74.53** | 79.31 | **77.17** | 74.46 |

Table 2: Accuracy under different White-box Attack attack on CIFAR-100 and SVHN

| MODELS | CIFAR-100($\epsilon = 8$) | | | | | SVHN($\epsilon = 8$) | | | | | |
|---|---|---|---|---|---|---|---|---|---|---|---|
| | CLEAN | FGSM | PGD20 | PGD100 | CW20 | CW100 | CLEAN | FGSM | PGD20 | PGD100 | CW20 | CW100 |
| STANDARD | **79.00** | 10.00 | 0.00 | 0.00 | 0.00 | 0.00 | **97.20** | 53.00 | 0.30 | 0.10 | 0.30 | 0.10 |
| AT | 59.90 | 28.50 | 22.60 | 22.30 | 23.20 | 23.00 | 93.90 | 68.40 | 47.90 | 46.00 | 48.70 | 47.30 |
| LAT | 60.94 | - | 27.03 | 26.41 | - | - | 60.94 | - | 27.03 | 26.41 | - | - |
| BILATERAL | 68.20 | 60.80 | 26.70 | 25.30 | - | 22.10 | 94.10 | 69.80 | 53.90 | 50.30 | - | 48.90 |
| FS | 73.90 | **61.00** | 47.20 | 46.20 | 34.60 | 30.60 | 96.20 | 83.50 | 62.90 | 52.00 | 61.30 | 50.80 |
| ATLD | 74.90 | 57.88 | 48.43 | 48.29 | 40.52 | 40.36 | 96.90 | **93.30** | 68.42 | 55.46 | 66.32 | 52.62 |
| ATLD-IMT | 63.17 | 56.64 | **53.26** | **53.35** | **50.80** | **49.77** | 85.85 | 89.07 | **79.93** | **74.70** | **79.44** | **74.59** |
| ATLD-IMT+ | 68.17 | 56.61 | 50.67 | 50.43 | 46.92 | 46.15 | 92.43 | 90.34 | 78.33 | 70.96 | 77.52 | 71.04 |

## 5.2 DEFENDING BLACK-BOX ATTACKS

To further verify the robustness of ATLD, we conduct transfer-based black-box attack experiments on CIFAR-10. More black-box attack results on CIFAR-100 and SVHN are listed in Appendix.

Four different models are used for generating test time attacks including the Vanilla Training model, the Adversarial Training with PGD model, the Feature Scattering Training model and our model. As demonstrated by the results in Table 3, our proposed approach can achieve competitive performance almost in all the cases. Specifically, ATLD outperforms Feature Scattering significantly in 8 cases while it demonstrates comparable or slightly worse accuracy in the other 3 cases.

It deserves our attention that ATLD-IMT appears to have a negative impact on the black-box attacks though it stills performs much better than PGD. This may be explained in several aspects. On one hand, the distributions of adversarial examples produced by different models may differ significantly in the latent space; on the other hand, our discriminator lacks the ability to deal with the unseen distributions since the discriminator only distinguishes one type of adversarial examples from the natural data during training. We will leave the investigation of this topic as future work.

Table 3: Accuracy under black-box attack on CIFAR-10

| DEFENSE MODELS | ATTACKED MODELS | | | | | | | | | | | |
| | VANILLA TRAINING | | | ADVERSARIAL TRAINING | | | FEATURE SCATTERING | | | OURS | | |
| | FGSM | PGD20 | CW20 | FGSM | PGD20 | CW20 | FGSM | PGD20 | CW20 | FGSM | PGD20 | CW20 |
| AT | 84.62 | 84.89 | 84.83 | 72.20 | 63.77 | 63.27 | 82.26 | 80.56 | 79.31 | 82.56 | 81.34 | 79.97 |
| FS | **88.64** | 89.25 | 89.31 | 77.18 | **66.59** | **66.40** | 82.80 | 81.01 | 78.34 | 82.99 | 81.56 | 79.92 |
| ATLD | 88.11 | **91.36** | **91.52** | **80.13** | 65.91 | 65.76 | **85.97** | **84.43** | **82.15** | **84.47** | **82.24** | **80.34** |
| ATLD-IMT | 83.40 | 86.17 | 86.41 | 73.66 | 61.59 | 61.62 | 84.37 | 81.18 | 78.18 | 83.68 | 81.05 | 77.95 |
| ATLD-IMT+ | 86.53 | 89.93 | 89.91 | 79.16 | 65.61 | 65.07 | 85.76 | 83.75 | 81.40 | 83.90 | 81.98 | 79.77 |

## 6 CONCLUSION

We have developed a novel adversarial training method which leverages both the local and global information to defend adversarial attacks in this paper. In contrast, existing adversarial training methods mainly generate adversarial perturbations in a local and supervised fashion, which could however limit the model's generalization. We have established our novel framework via an adversarial game between a discriminator and a classifier: the discriminator is learned to differentiate globally the latent distributions of the natural data and the perturbed counterpart, while the classifier is trained to recognize accurately the perturbed examples as well as enforcing the invariance between the two latent distributions. Extensive empirical evaluations have shown the effectiveness of our proposed model when compared with the recent state-of-the-art in defending adversarial attacks in both the white-box and black-box settings.

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

## APPENDIX

## A  LIST OF MAJOR NOTATION

For clarity, we list the major notations that are used in our model.

- $X_{org} = \{x : x \sim Q_0\}$: the set of clean data samples, where $Q_0$ is its underlying distribution;

- $X_p = \{x' : x' \in B(x, \epsilon), \ \forall x \sim Q_0\}$: the set of perturbed samples, the element $x' \in X_p$ is in the $\epsilon$-neighborhood of the clean example $x \sim Q_0$;

- $f_\theta$: the mapping function from input to the latent features of the last hidden layer (i.e., the layer before the softmax layer);

- $Q_\theta$: the underlying distribution of the latent feature $f_\theta(x)$ for all $x \in X_{org}$;

- $P_\theta$: the underlying distribution of the latent feature $f_\theta(x')$ for all $x' \in X_p$;

- $\mathcal{P}$: the feasible region of the latent distribution $P_\theta$, which is defined as $\mathcal{P} \triangleq \{P : f_\theta(x') \sim P$ subject to $\forall x \sim Q_0, x' \in B(x, \epsilon)\}$.

- $X_{adv}$: the set of the worst perturbed samples or manifold adversarial examples, the element $x^{adv} \in X_{adv}$ are in the $\epsilon$-neighborhood of clean example $x \sim Q_0$;

- $P_\theta^*$: the worst latent distribution within the feasible region $\mathcal{P}$ which leads to the largest divergence or the underlying distribution of the latent feature $f_\theta(x^{adv})$ for all $x^{adv} \in X_{adv}$;

## B  ADDITIONAL EXPERIMENT DETAILS

### B.1  MODEL ROBUSTNESS AGAINST PGD AND CW ATTACKER UNDER DIFFERENT ATTACK BUDGETS

We further evaluate the model robustness against PGD and CW attacks under different attack budgets with a fixed attack step of 20. These results are shown in Figure 3. It is observed that the performance of Adversarial Training with the PGD method (AT) drops quickly as the attack budget increases. The Feature Scattering method (FS) can improve the model robustness across a wide range of attack budgets. The proposed approach ADLT-IMT further boosts the performance over Feature Scattering by a large margin under different attack budgets especially under CW attack, except that our ADLT-IMT is slightly inferior to Feature Scattering under PGD attack with budget $\epsilon = 20$ on CIFAR-10.

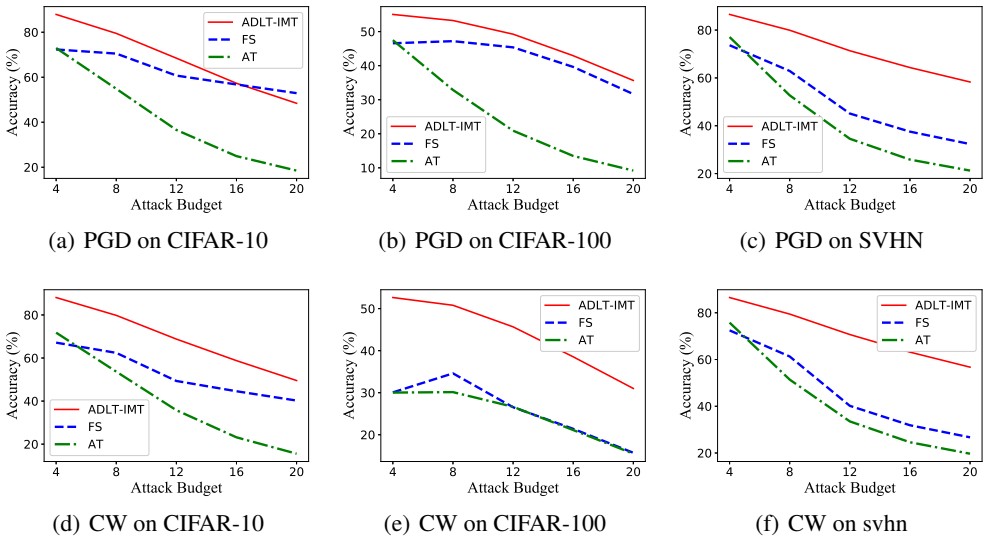

Figure 3: Model performance under PGD and CW attacks with different attack budgets.

## B.2 MODEL ROBUSTNESS AGAINST AUTOATTACK AND RAYS

As shown in (Croce & Hein, 2020; Chen & Gu, 2020), several models (such as Feature Scattering) could achieve high enough robustness against PGD and CW attack, but they may fail to defend more stronger attacks. To further evaluate the model robustness against stronger attacks, we evaluate the robustness of our proposed method IMT+ against AutoAttack (Croce & Hein, 2020) and RayS (Chen & Gu, 2020) attacks with $L_\infty$ budget $\epsilon = 8$ on CIFAR-10 and CIFAR-100.

We first compare the accuracy of the proposed ATLD-IMT+ with several competitive methods on CIFAR-10 in Table 4 to defend the AutoAttack (AA) and Rays attacks, including: (1) Traditional Adversarial Training with PGD (AT) (Madry et al., 2017); (2) TRADES: trading adversarial robustness off against accuracy (Zhang et al., 2019); (3) Feature Scattering: generating adversarial examples with considering inter-relationship of samples (Zhang & Wang, 2019); (4) Robust-overfitting: improving models adversarial robustness by simply using early stop (Rice et al., 2020); (5) Pretraining: improving models adversarial robustness with pre-training (Hendrycks et al., 2019); (6)WAR: mitigating the perturbation stability deterioration on wider models (Wu et al., 2020); (7) RTS: achieving high robust accuracy with semisupervised learning procedure (self-training) (Carmon et al., 2019); (8) Gowal et al. (2020): achieving state-of-the-art results by combining larger models, Swish/SiLU activations and model weight averaging.

These comparison algorithms attain the most competitive performance on defending AA attack. As observed, overall, our proposed method achieves a clear superiority over all the defence approaches on both the clean data and adversarial examples (except on clean data, ours is slightly inferior to Gowal et al. (2020) which is however trained with additional data). Note that Pretraining, WAR and Gowal et al. (2020) with footnote require additional data for training (e.g. unlabeled data, pre-training).

We also report the accuracy of ATLD-IMT+ with the state-of-the-arts methods on CIFAR-100 in Table 5 against the AutoAttack (AA). Our proposed method again achieves on both the clean data and AA attacked examples significant better performance than all the other defense approaches (without data augmentation). Furthermore, it is noted that, while our ATLD-IMT+ method is just slightly inferior to Gowal et al. (2020) (which is trained with additional data), it is substantially ahead of the normal version of Gowal et al. (2020).

Table 4: Accuracy under AutoAttack (AA) and RayS on CIFAR-10

| MODELS | CIFAR-10($\epsilon = 8$) | | | |
| --- | --- | --- | --- | --- |
| | CLEAN | CW100 | AA | RayS |
| AT | 87.14 | 50.60 | 44.04 | 50.70 |
| TRADES | 84.92 | 56.43 | 53.08 | 57.30 |
| FS | 89.98 | 60.6 | 36.64 | 44.50 |
| ROBUST-OVERFITTING | 85.34 | 58.00 | 53.42 | 58.60 |
| PRETRAINING[*] | 87.11 | 57.40 | 54.92 | 60.10 |
| WAR[*] | 85.60 | 59.78 | 59.78 | 63.2 |
| RTS[*] | 89.69 | 62.50 | 59.53 | 64.6 |
| GOWAL ET AL. (2020) | 85.29 | 57.14 | 57.20 | - |
| GOWAL ET AL. (2020)[*] | **91.10** | 65.87 | 65.88 | - |
| ATLD-IMT | 86.42 | **75.20** | 70.49 | 70.60 |
| ATLD-IMT+ | 90.78 | 74.46 | **70.60** | **81.68** |

[*] INDICATES MODELS WHICH REQUIRES ADDITIONAL DATA FOR TRAIN-ING.

Table 5: Accuracy under AutoAttack (AA) on CIFAR-100

| MODELS | CIFAR-100($\epsilon = 8$) | | | |
| --- | --- | --- | --- | --- |
| | CLEAN | CW100 | AA | RayS |
| ROBUST-OVERFITTING | 53.83 | 28.10 | 18.95 | - |
| PRETRAINING[*] | 59.23 | 33.50 | 28.42 | - |
| GOWAL ET AL. (2020) | 60.86 | 30.67 | 30.03 | - |
| GOWAL ET AL. (2020)[*] | **69.15** | 37.70 | **36.88** | - |
| ATLD-IMT | 63.17 | **49.77** | 31.09 | 41.98 |
| ATLD-IMT+ | 68.17 | 46.15 | 32.36 | **43.91** |

[*] INDICATES MODELS WHICH REQUIRES ADDITIONAL DATA FOR TRAIN-ING.

## B.3 BLACK-BOX RESULTS ON SVHN AND CIFAR-100

We conduct more evaluations on the transfer-based black-box attacks on SVHN and CIFAR-100. We report the results in Table 6. It can be observed that our proposed method overall outperforms Feature Scattering in most of the cases on SVHN. Surprisingly, the Adversarial Training method, i.e. the PGD, performs better than our method and Feature Scattering method in three cases. This also partially reveals the more challenging nature of defending black-box attacks than white-box attacks.

On CIFAR-100, it can be observed that our method and Feature Scattering are comparable. The performance of these two methods differs little though our method outperforms Feature Scattering significantly under PGD20 and CW20 against adversarial attacks generated from the Feature Scattering model.

Overall, though the proposed ATLD method may not lead to remarkably higher performance than the current state-of-the-art algorithms in defending black-box attacks (as we observed in the case of white-box attacks), it still generates overall better or comparable performance. We will again leave the further exploration of defending black-box attacks as our future work.

## B.4 ILLUSTRATION OF THE OVERLAID BOUNDARY CHANGE OF DIFFERENT METHODS

We conduct a toy example in Figure 4 to illustrate the effect on how the various methods would affect the decision boundary after the adversarial training is applied. In Figure 4, (a) shows the decision boundary trained with clean data; (b) shows the decision boundary adversarially trained with the perturbed samples by PGD; (c) presents the decision boundary given by the adversarial training of Feature Scattering; and (d) illustrates the decision boundary trained from our proposed ATLD. Clearly, both the PGD (Figure 4(b)) and the FS (Figure 4(c)) vary the original decision boundary significantly. Moreover, it can be observed that the adversarial training with PGD corrupts the data manifold completely. On the other hand, FS appears able to retain partially the data manifold information since it considers the inter-sample relationship locally. Nonetheless, its decision boundary appears non-smooth, which may hence degrade the performance. In contrast, as shown in Figure 4(d), our proposed method considers to retain the data manifold globally, which varies the

Table 6: Accuracy under black-box attack on SVHN and CIFAR-100

| DEFENSE MODELS | ATTACKED MODELS (SVHN) | | | | | | | | | | | |
| --- | --- | --- | --- | --- | --- | --- | --- | --- | --- | --- | --- | --- |
| | VANILLA TRAINING | | | ADVERSARIAL TRAINING | | | FEATURE SCATTERING | | | OURS | | |
| | FGSM | PGD20 | CW20 | FGSM | PGD20 | CW20 | FGSM | PGD20 | CW20 | FGSM | PGD20 | CW20 |
| AT | **88.86** | 89.17 | 89.11 | 75.56 | 62.54 | 62.74 | 90.76 | 88.62 | 88.76 | 90.98 | **90.10** | **90.31** |
| FS | 80.89 | 86.65 | 87.10 | 82.27 | 65.59 | 65.18 | **96.59** | 81.91 | 82.52 | **93.11** | 81.65 | 82.46 |
| ATLD | 82.61 | **89.39** | **89.49** | **83.57** | **66.00** | 65.28 | 96.42 | **91.88** | **92.36** | 91.75 | 86.46 | 87.37 |
| ATLD-IMT | 76.71 | 75.27 | 75.51 | 76.20 | 61.57 | 62.91 | 86.10 | 81.67 | 81.60 | 87.85 | 83.60 | 83.59 |
| ATLD-IMT+ | 80.27 | 82.18 | 82.53 | 79.61 | 64.63 | 64.84 | 90.33 | 86.84 | 82.06 | 87.97 | 83.27 | 83.71 |
| DEFENSE MODELS | ATTACKED MODELS (CIFAR-100) | | | | | | | | | | | |
| | VANILLA TRAINING | | | ADVERSARIAL TRAINING | | | FEATURE SCATTERING | | | OURS | | |
| | FGSM | PGD20 | CW20 | FGSM | PGD20 | CW20 | FGSM | PGD20 | CW20 | FGSM | PGD20 | CW20 |
| AT | 60.46 | 60.88 | 60.92 | 47.16 | 40.37 | 39.53 | 61.61 | 61.20 | 60.78 | 60.71 | 60.99 | 60.96 |
| FS | 66.51 | **71.47** | **71.88** | **56.96** | 44.91 | **46.37** | **74.00** | 62.07 | 62.07 | **69.35** | 63.36 | 63.50 |
| ATLD | **67.30** | 71.42 | 71.56 | 56.01 | **44.92** | 45.84 | 71.52 | **67.72** | **68.32** | 67.20 | 63.02 | **63.69** |
| ATLD-IMT | 54.05 | 57.01 | 58.30 | 45.51 | 37.21 | 39.61 | 60.29 | 59.72 | 57.93 | 59.06 | 59.50 | 59.02 |
| ATLD-IMT+ | 57.77 | 61.40 | 60.66 | 48.15 | 39.07 | 41.30 | 63.09 | 61.48 | 60.78 | 61.32 | 60.95 | 59.93 |

decision boundary slightly. This may explain why our proposed ATLD method could outperform the other approaches.

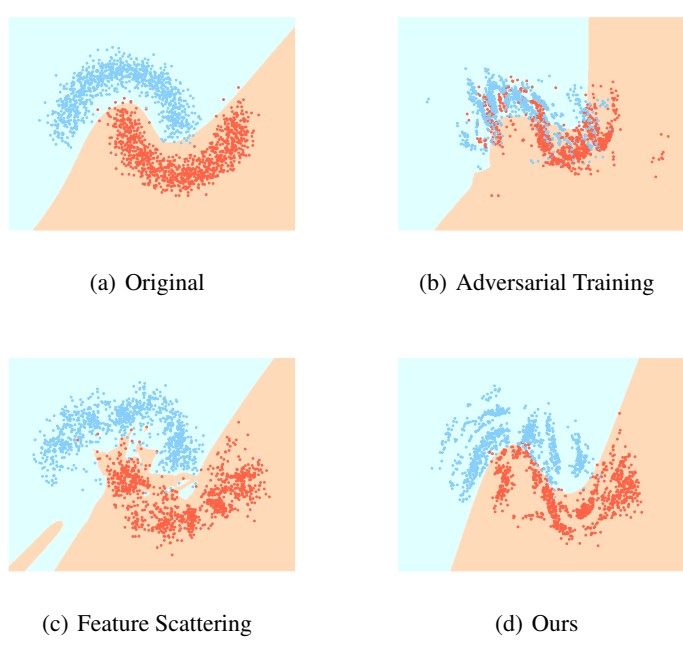

(a) Original

(b) Adversarial Training

(c) Feature Scattering

(d) Ours

Figure 4: The overlaid decision boundary after the various adversarial training is applied

### B.5 FURTHER DETAILS OF ATLD-IMT

We elaborate the training procedure of our IMT in this section. The overall architecture of ATLD-IMT is plotted in Figure 5. A test sample $x$ is fed into the classifier, and the discriminator outputs the prediction. A special perturbation in IMT is then computed from the loss $D_W$ and added back to $x$; in this way, the sample would be pushed towards the manifold of natural samples, which is supposed to be further away from the decision boundary. The prediction of the transformed $x^t$ by the adversarially-trained classifier will then be output as the label of $x$.

To illustrate clearly the effect of our ATLD-IMT, we conduct additional toy experiments as shown in Figure 6 where we respectively plot the clean or natural data, perturbed data attacked by PGD, and adjusted data by ATLD-IMT in (a), (b), and (c). Moreover, the decision boundary is given by ATLD

in all the three sub-figures. In (a), it deserves our attention that the boundary learned by ATLD could classify natural data well compared to the PGD and Feature Scattering as shown in Section A.3. As observed in (b), the perturbations generated by PGD will push the natural samples toward or even cross the decision boundary. Our proposed IMT can push the samples towards the manifold of natural examples as observed in (c). Since the manifold of natural examples would be more separable, this may further increase the classification performance as observed in the experiments.

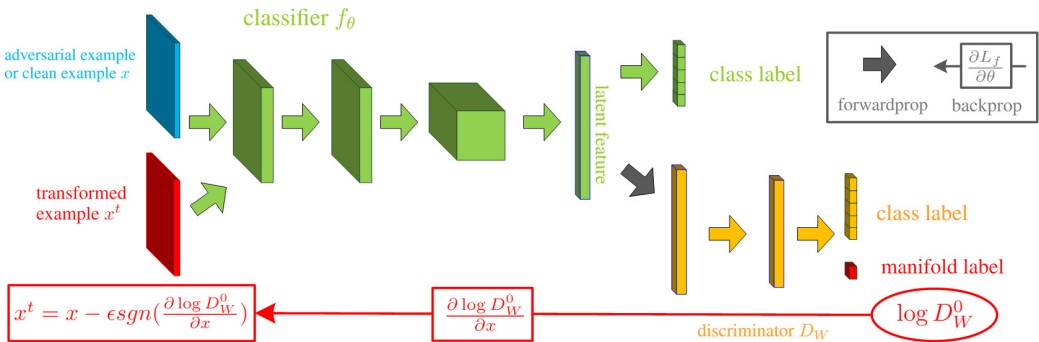

Figure 5: Detailed Procedure of IMT. 1) The natural example or adversarial example $x$ is fed into the network, and the discriminator outputs its prediction. The loss $\log D_W$ is computed and the transformed example $x^t$ (red arrow) is then generated. 2) The transformed sample is fed into the network and classified by the adversarially-trained network.

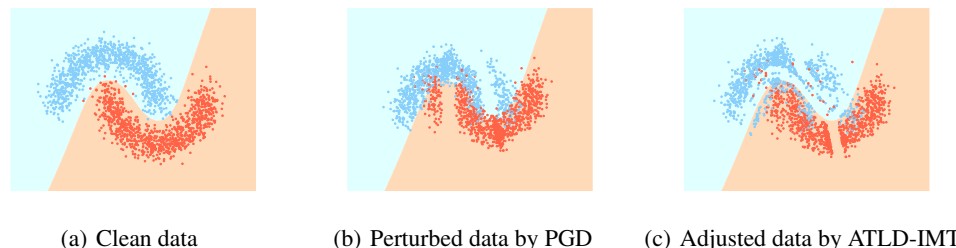

(a) Clean data      (b) Perturbed data by PGD      (c) Adjusted data by ATLD-IMT

Figure 6: Illustration of ATLD-IMT. The decision boundary is given by ATLD in all the three sub-figures, while (a) shows clean data, (b) draws perturbed data attacked by PGD, and (c) plots adjusted data by ATLD-IMT. Our proposed IMT can push the samples towards the manifold of natural examples as observed in (c). Since the manifold of natural examples would be more separable, this may further increase the classification performance.

### B.6 ILLUSTRATION OF VECTOR FIELD OF DIFFERENT PERTURBATION SCHEMES

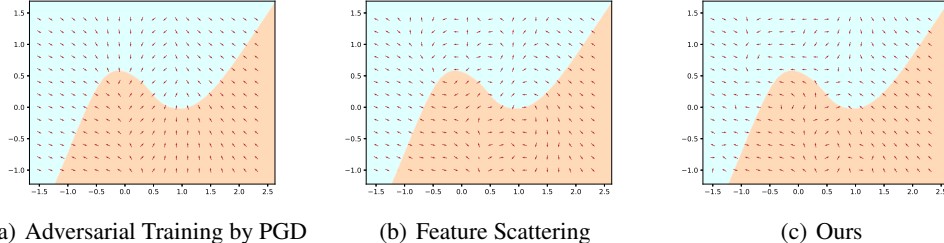

(a) Adversarial Training by PGD      (b) Feature Scattering      (c) Ours

Figure 7: Vector field illustration of different perturbation schemes. (a) PGD, (b) Feature Scattering, (c) the proposed ATLD method. The overlaid boundary is from the model trained on clean data. The figure plots the direction of adversarial perturbations at different points. It is worth noting that most directions of the adversarial perturbations for the conventional adversarial training methods point to the decision boundary. It indicates that the resulting adversarial examples are easily biased towards the decision boundary which potentially corrupts the structure of the underlying distribution. The perturbation directions for Feature Scattering and our method are influenced by decision boundary less.

## C   DETAILED DERIVATION

In this section, we provide the details about the derivation for the main objective function (6) and elaborate how to compute the adversarial examples and the transformed examples.

### C.1   DERIVATION FOR MAIN OBJECTIVE FUNCTION (6)

We start with minimizing the largest $f$-divergence between latent distributions $P_\theta$ and $Q_\theta$ induced by perturbed example $x'$ and natural example $x$. And we denote their corresponding probability density functions as $p(z)$ and $q(z)$. According to Eq. (3), we have

$$
\begin{aligned}
\min_\theta \max_{Q_\theta} D_f(P_\theta \| Q_\theta) &= \min_\theta \max_{q(z)} \int_{\mathcal{Z}} q(z) \sup_{t \in \mathrm{dom}\, f^*} \{ t \frac{p(z)}{q(z)} - f^*(t) \} dx \\
&\geq \min_\theta \max_{q(z)} \sup_{T \in \tau} ( \int_{\mathcal{Z}} p(z) T(z) dz - \int_{\mathcal{Z}} q(z) f^*(T(z)) dz ) \\
&= \min_\theta \max_{Q_\theta} \sup_W \left\{ \mathbb{E}_{z \sim P_\theta}[g_f(V_W(z))] + \mathbb{E}_{z \sim Q_\theta}[-f^*(g_f(V_W(z)))] \right\} \\
&= \min_\theta \sup_W \left\{ \mathbb{E}_{x \sim \mathcal{D}} \{ \max_{x' \in B(x,\epsilon)} [g_f(V_W(f_\theta(x')))] + [-f^*(g_f(V_W(f_\theta(x))))] \} \right\}
\end{aligned}
\tag{9}
$$

To compute the Jensen-Shannon divergence between $P_\theta$ and $Q_\theta$, we set $g_f(t) = -\log(1 + e^{-t})$ and $f^*(g) = -\log(2 - e^g)$. Then, we have

$$
\min_\theta \max_{Q_\theta} D_{JS}(P_\theta \| Q_\theta) \geq \min_\theta \sup_W \left\{ \mathbb{E}_{x \sim \mathcal{D}} \{ \max_{x' \in B(x,\epsilon)} [\log D_W(f_\theta(x'))] + [1 - \log D_W(f_\theta(x))] \} \right\}
\tag{10}
$$

where $D_W(x) = 1/(1 + e^{-V_W(x)})$. (10) is equivalent to optimize the lower bound of Jensen-Shannon divergence between $P_\theta$ and $Q_\theta$. With disentangling the computation of adversarial examples from Eq. (10) and further considering the classification loss for the classifier $L_f$ and the

discriminator $L_d^{1:C}$, we can obtain the final objective:

$$\min_\theta \Big\{ \sup_W \sum_{i=1}^N \underbrace{[\log D_W^0(f_\theta(x_i^{adv})) + (1 - \log D_W^0(f_\theta(x_i)))]}_{L_d^0}$$

$$+ \underbrace{L(x_i^{adv}, y_i; \theta)}_{L_f} + \min_W \underbrace{[l(D_W^{1:C}(f_\theta(x_i)), y_i) + l(D_W^{1:C}(f_\theta(x_i^{adv})), y_i)]}_{L_d^{1:C}} \Big\},$$

(11)

$$\text{s.t.} \quad x_i^{adv} = \arg \max_{x_i' \in B(x_i, \epsilon)} \underbrace{[\log D_W^0(f_\theta(x_i')) + (1 - \log D_W^0(f_\theta(x_i)))]}_{L_d^0}$$

### C.2 COMPUTATION FOR ADVERSARIAL EXAMPLE AND TRANSFORMED EXAMPLE

To compute the adversarial example, we need to solve the following problem:

$$x_i^{adv} = \arg \max_{x_i' \in B(x_i, \epsilon)} \underbrace{[\log D_W^0(f_\theta(x_i')) + (1 - \log D_W^0(f_\theta(x_i)))]}_{L_d^0}$$

(12)

It can be reformulated as computing the adversarial perturbation as follows:

$$r_i^{adv} = \arg \max_{\|r\|_\infty \leq \epsilon} [L_d^0(x_i + r_i, \theta)]$$

(13)

We first consider the more general case $\|r\|_p \leq \epsilon$ and expand (13) with the first order Taylor expansion as follows:

$$r_i^{adv} = \arg \max_{\|r\|_p \leq \epsilon} [L_d^0(x_i, \theta)] + \nabla_x \mathcal{F}^T r_i$$

(14)

where $\mathcal{F} = L(x_i, \theta)$. The problem (14) can be reduced to:

$$\max_{\|r_i\|_p = \epsilon} \nabla_x \mathcal{F}^T r_i$$

(15)

We solve it with the Lagrangian multiplier method and we have

$$\nabla_x \mathcal{F} r_i = \lambda(\|r_i\|_p - \epsilon)$$

(16)

Then we make the first derivative with respect to $r_i$:

$$\nabla_x \mathcal{F} = \lambda \frac{r_i^{p-1}}{p(\sum_j (r_i^j)^p)^{1 - \frac{1}{p}}}$$

(17)

$$\nabla_x \mathcal{F} = \frac{\lambda}{p} (\frac{r_i}{\epsilon})^{p-1}$$

$$(\nabla_x \mathcal{F})^{\frac{p}{p-1}} = (\frac{\lambda}{p})^{\frac{p}{p-1}} (\frac{r_i}{\epsilon})^p$$

(18)

If we sum over two sides, we have

$$\sum (\nabla_x \mathcal{F})^{\frac{p}{p-1}} = \sum (\frac{\lambda}{p})^{\frac{p}{p-1}} (\frac{r_i}{\epsilon})^p$$

(19)

$$\|\nabla_x \mathcal{F}\|_{p^*}^{p^*} = (\frac{\lambda}{p})^{p^*} * 1$$

(20)

where $p^*$ is the dual of $p$, i.e. $\frac{1}{p} + \frac{1}{p^*} = 1$. We have

$$(\frac{\lambda}{p}) = \|\nabla_x \mathcal{F}\|_{p^*}$$

(21)

By combining (18) and (21), we have

$$
\begin{aligned}
r_i^* &= \epsilon \mathrm{sgn}(\nabla_x \mathcal{F})\Big(\frac{|\nabla_x \mathcal{F}|}{\|\nabla_x \mathcal{F}\|_{p^*}}\Big)^{\frac{1}{p-1}} \\
&= \epsilon \mathrm{sgn}(\nabla_x L_d^0)\Big(\frac{|\nabla_x L_d^0|}{\|\nabla_x L_d^0\|_{p^*}}\Big)^{\frac{1}{p-1}}
\end{aligned}
\tag{22}
$$

In this paper, we set $p$ to $\infty$. Then we have

$$
\begin{aligned}
r_i^* &= \epsilon \lim_{p \to \infty} sgn(\nabla_x L_d^0)\Big(\frac{|\nabla_x L_d^0|}{\|\nabla_x L_d^0\|_{p^*}}\Big)^{\frac{1}{p-1}} \\
&= \epsilon \mathrm{sgn}(\nabla_x L_d^0)\Big(\frac{|\nabla_x L_d^0|}{\|\nabla_x L_d^0\|_1}\Big)^0 \\
&= \epsilon \mathrm{sgn}(\nabla_x L_d^0)
\end{aligned}
\tag{23}
$$

Then we can obtain the adversarial example:

$$
x_i^* = x_i + \epsilon \mathrm{sgn}(\nabla_x L_d^0)
\tag{24}
$$

To compute the transformed example, we need to solve the following problem:

$$
r^* = \arg \min_{\|r\|_\infty \leq \epsilon} \log D_W^0(f_\theta(x + r)).
\tag{25}
$$

With the similar method, we can easily get the transformed example $x^t$

$$
x^t = x - \epsilon \mathrm{sgn}(\nabla_x \log D_W^0).
\tag{26}
$$

