# OpenReview forum: "Improving Model Robustness with Latent Distribution Locally and Globally"
_ICLR.cc/2021/Conference — Reject_

### Official Review · AnonReviewer1 · 2020-10-29
**Needs better technical exposition**

**Rating:** 4
**Confidence:** 4

**Review:**

This paper presents a framework for adversarial robustness via incorporating local and global structure of the data manifold. Specifically, the key motivation is that standard adversarial methods typically use only sample specific perturbations for generating the adversarial examples, and thus using them for robustness of the learning model is limited. Instead the paper proposes to capture the global data manifold as well in the robustifying framework. To this end, an objective is presented (4,5) that uses latent data distributions, with the goal that the adversarial perturbations should maximize the f-divergence against the latent distribution of the clean samples. Experiments are provided on several datasets and demonstrate significant performance improvements.

Pros:
1. The key idea of using the global data manifold into the robustifying framework is quite interesting.
2. Experiments demonstrate good empirical benefits of the approach.

Cons:
1. While, the paper seemed well organized in the beginning, I got totally lost with Eq. (4-5). As I see, this objective is inaccurate and needs significant refinement. Specifically, it is unclear how is x^{adv} is related to x, and how is x^{adv} related to P*_theta? The paper tries to explain this objective in the paragraph below, but the explanation is very confusing as well.  A few other things that could help here:
a) It is said that "Q_theta and P_\theta* are the latent distributions induced by the natural example x". How can a single data point induce a distribution? Do you assume the feature map from a hidden layer of a network represents a distribution? If so, in what sense?
b) "The adversarial example is crafted to induce the worst case distribution P*". How is it crafted and what is the relation between P* and x? This is the key connection that is missing from (4-5).

2. Moving along, Section 4.1 is organized very poorly as well. I believe too many concepts are tied together into one formulation in (6), making it hard to decipher. For example, why to include the classifier D^{1:C} within this formulation? Why not talk about it elsewhere and focus on the meat of the objective, systematically?

3. Further, as I understand, x^{adv} is the first step that happens in (6), however, there is no "adversary" in this case, instead is finding a perturbed sample x' that maximizes the f-divergence. In what sense is x^{adv} then an adversarial sample? Perhaps the paper should re-define what is the definition of an adversarial example that it is using, to clearly state what the idea is. Technically, there is no requirement that the point x^{adv} found by this step will promote any data misclassification; however can be any point that is within a B(x,\eps) ball from x, and that happens to maximize this divergence loss. Note that none of the other components D_W, f_theta, etc. are well trained in doing this optimization. So they could also be sub-optimal (in the sense of what the paper argues in the beginning of Page 4).

4. Why is the middle formula in (6) minimizing over W to have both x and x^adv matched with the same label? Again, where is the adversary here? Or for that matter, how will the proposed approach achieve adversarial robustness ?

Minor comments:
a. What is \tau and T in (3)?
b. How is f_\theta defined in (6)?
c. The paper writes that back and forth that there is no use of label information in the setup, however has labels used in discriminator in (6). This is very confusing.
d. There is also reference to data manifold and manifold label in Figure 2, but these are not clearly explained. What precisely is the data manifold? Is it the latent distribution for a specific label?
e. Page 4, top para: "without considering the inter-relationship between data samples". Won't this relation be captured implicitly through the neural network parameters theta when perturbations on all the samples are used in the training process?

Overall, I think this paper needs a thorough revision to explain well its technical contributions.

---

> ### Author Response · Authors · 2020-11-25
> **Response to AnonReviewer1 (Part 1/3)**
>
> We appreciate R1 for the constructive comments which truly help us improve the quality
> of the paper. We believe the main concerns from R1 lie at the confusing writing in the original paper,
> which however may not affect the correctness of the proposed theory as well as its significance.
> Following the valuable suggestions, we have updated our paper by making clarifications and revising
> the descriptions of the equations and some texts to avoid potential confusion.
> 1. Response to Cons1:
> According to R1’s comments, we have made Eqs.(4)-(5) more explicitly and clarified the
> reviewer’s questions in the revised version (in blue).
> For clarity, we also list the major definitions in Appendix A of the updated paper. We
> provide these definitions as follows for convenience.
>
>     * $X_{org}=$ {${x:x\sim Q_0}$}: the set of clean data samples, where $Q_0$ is its underlying
> distribution;
>
>     * $X_{p}=$ {${x': x'\in B(x,\epsilon), \forall x\sim Q_0}$}: the set of perturbed samples, the element $x'\in X_{p}$ is in the $\epsilon$-neighborhood of the clean example $x\sim Q_0$;
>
>     * $f_\theta$: the mapping function from input to the latent features of the last hidden layer (i.e., the layer before the softmax layer);
>
>     * $Q_\theta$: the underlying distribution of the latent feature $f_\theta (x)$ for all $x \in X_{org}$;
>
>     * $P_\theta$: the underlying distribution of the latent feature $f_\theta (x')$ for all $x'\in X_{p}$;
>
>     * $\mathcal{P}$: the feasible region of the latent distribution $P_{\theta}$, which is defined as $\mathcal{P} \triangleq $ {${P:f_\theta(x')\sim P \text{ subject to } \forall x\sim Q_0, x'\in B(x,\epsilon)}$}.
>
>      * $X_{adv}$: the set of the worst perturbed samples or manifold adversarial examples, the element $x^{adv}\in X_{adv}$ are in the $\epsilon$-neighborhood of clean example $x\sim Q_0$;
>
>     * $P_\theta^*$: the worst latent distribution within the feasible region $\mathcal{P}$ which leads to the largest divergence or the underlying distribution of the latent feature $f_\theta (x^{adv})$ for all $x^{adv}\in X_{adv}$;
>
>
>    While the optimization problem in Eqs.(4)-(5) remains equivalent to the previous formulation, we introduce some parameters and definitions to explicitly indicate the relation of the clean examples $X_{org}$, the manifold adversarial examples $X_{adv}$, and the latent distributions $P_{\theta}$ and $Q_{\theta}$ in the last hidden layer.
>
>   First of all, we would clarify that our aim is to enhance the distributional robustness in an unsupervised fashion, and the resulting adversarial examples $X_{adv}$ are not the same as the conventional definition. We refer to them as the manifold adversarial examples, which are drawn from an underlying distribution perturbed from the underlying distribution of the input samples $X_{org}$.
>
>   For distributional robustness, the initial objective is to figure out the underlying distribution of the manifold adversarial examples, which is the worst perturbation of $Q_0$, so that latent distribution $P_{\theta}$ is as apart from $Q_{\theta}$ as possible through the $f$-divergence metric.
>
>   Since this is in general intractable, we instead aim to maximize the $f$-divergence between two latent distributions, subject to the constraint that $P_{\theta} \in \mathcal{P}$. It is expected that the underlying distribution of $X_{adv}$ that yields the worst-case latent distribution $P_{\theta}^{*}$ could be utilized  to enhance robustness through adversarial training.
>
>   To directly answer the reviewer's question, in Eqs.(4)-(5) of the revised version, the adversarial example $x^{adv}$ is related to $x$ and $P_\theta$ through the feasible region $\mathcal{P}$.

---

> ### Author Response · Authors · 2020-11-25
> **Response to AnonReviewer1 (Part 2/3)**
>
> 2. Response to Cons2:
> We highly appreciate this constructive suggestion. Following the reviewer's suggestion, we have re-organized Section 4.1 and split Eq.(6) in the previous version to two equations Eq. (6)-(7) in the new submission (in blue).
>
>   Specifically, we made a direct translation from Eq. (4)-(5) to Eq. (6) in the revised version with the term $L^{1:C}_d$ relegated to Eq. (7).
>
>   In the revised version, we explained how Eq. (6) comes from Eqs.(4)-(5) and described each term in Eq. (6). Specifically, we approximate Jensen-Shannon divergence between $P_\theta^*$($P_\theta$) and $Q_\theta$ with $\sup_W\sum_{i=1}^NL_d$ and we minimize the classification loss on adversarial examples by minimizing $\sum_{i=1}^NL_f$. The optimization problem (6) is solved by alternatively updating parameters $\theta$ and $W$ and crafting adversarial examples $\{x_i^{adv}\}_{i=0}^N$. Although $D_w$ cannot measure the divergence between two latent distributions exactly at the first several training steps, when the parameters $W$ converge, $D_W$ can help evaluate the divergence between distributions induced by perturbed examples and clean ones. Then the worst perturbed examples can be crafted with the help of the discriminator $D_W$ as the constraint of (6).
>
>   Moreover, we detailed why the term $L^{1:C}_d$ comes into play in Eq. (7). It is a regularization term inspired by some existing works to deal with the failure of the discriminator networks. Several work [1][2][3] reveals high risk of failure in measuring only a fraction of components underlying different distributions with the discriminator networks, and shows that even if the discriminator is fully confused, there is no theoretical guarantee that two different distributions are identical. To alleviate such problem, we additionally train the discriminator $D_W$ to predict the class labels for latent features as [4][5] via adding the term $L^{1:C}_d$.
>
>   [1]Martin Arjovsky and L`eon Bottou. Towards principled methods for training generative
> adversarial networks. In Stat, volume 1050, 2017.
>
>   [2]Tong Che, Yanran Li, Athul Paul Jacob, Yoshua Bengio, andWenjie Li. Mode regularized
> generative adversarial networks. 2016.
>
>   [3]Tengyu Ma. Generalization and equilibrium in generative adversarial nets (gans) (invited
> talk). In the 50th Annual ACM SIGACT Symposium, 2018.
>
>   [4]Augustus Odena, Christopher Olah, and Jonathon Shlens. Conditional image synthesis
> with auxiliary classifier gans. In International conference on machine learning, pp.
> 2642–2651, 2017.
>
>   [5]Mingsheng Long, Zhangjie Cao, Jianmin Wang, and Michael I Jordan. Conditional
> adversarial domain adaptation. In Advances in Neural Information Processing Systems,
> pp. 1640–1650, 2018.
>
>
> 3. Response to Cons3:
> We thank the reviewer for raising this point. In general, we agree with the reviewer that the definition of adversarial examples here is different from that of the conventional one. It is because the adversary is introduced in an unsupervised manner (i.e., no label information is required to produce the adversarial examples), and thus the adversarial examples do not necessarily result in a mis-classification.
>
>    Following the reviewer's suggestions, we emphasize the difference of 'manifold' adversarial examples from the conventional ones in the revised paper. In particular, the 'manifold' adversarial examples in this paper are defined as the worst perturbed examples which induce the most different latent features (latent distribution) from the clean ones (measured by JS-divergence). When the different latent representations lead to different outputs or predictions, such adversarial examples can result in a mis-classification and agree with the definition of the conventional adversarial examples.
>
>   We agree with the reviewer that the solution to the optimisation problem in Eq. (6) could be sub-optimal due to not well-trained neural networks. However, we argue that the optimal solution might be unnecessary with respect to adversarial robustness. Our method aims to promote the distributional robustness by enforcing the invariance between the latent distributions of adversarial examples and clean ones. Even if our method may not generate the strongest adversarial examples (to fool the classifier) as the traditional supervised method, in practice our method can still generate adversarial examples near the classification boundary (as shown in Figure 1) to help boost the model robustness. Moreover, compared with traditional methods, our method can retain more structure information of the distribution as Figure 1 shows. It is also noted that although the discriminator $D_W$ cannot help generate the optimal adversarial examples at the first several training steps, when the parameters $W$ of the discriminator converge, it can help generate the near-optimal adversarial examples.

---

> ### Author Response · Authors · 2020-11-25
> **Response to AnonReviewer1 (Part 3/3)**
>
> 4. Response to Cons4:
> The mentioned term is now in Eq. (7) in the revised version, which has both $x$ and $x_{adv}$ matched with the same class label. This is a regularization term to reduce the risk of failure of the discriminator networks. It has nothing to do with adversarial robustness. The aim is to help the discriminator to distinguish between the two distributions more accurately.
>
>   For the adversarial robustness, as mentioned in '`Response to Cons3', it can be achieved by the interplay between all terms in Eq. (7) that form a minimax game among the classifier, the discriminator and the adversarial examples. The adversarial examples are crafted to induce the most different latent features from the latent features of the clean data (with the help of the discriminator), while the classifier is trained to classify the adversarial examples generated by our proposed method as accurately as possible and enforce the invariance between the latent distributions of adversarial examples and clean ones.  By the outer minimization, the latent representations of such adversarial examples are enforced to be similar to clean ones. It means that the outputs or predictions of adversarial examples are similar to those of the cleaned ones. In other words, the DNN can classify such adversarial examples as accurately as clean ones, meaning that the adversarial robustness is guaranteed.
>
>
> 5. Response to Minor comments:
>
>     (a) $\tau$ is an arbitrary class of functions $T:\mathcal{X}\to \mathbb{R}$, we have added the description in the revised version.
>
>     (b) $f_\theta$ represents the latent features of the last hidden layer. We clarify this point in our revised version, for example, ''$f_\theta(x')$ and $f_\theta(x^{adv})$ represents the latent features of the perturbed example $x'$ and adversarial example $x^{adv}$ respectively''; ''$f_\theta(x_i)$ is the latent feature of the clean sample $x_i$''.
>
>     (c) Thank you for pointing out this issue. We remark that our proposed 'adversarial examples' are in a general sense, which are crafted by an adversary in an unsupervised manner. This is shown in the constraint in Eq.(6)-(7) in our revised paper.  It is different from the conventional definition, as it unnecessarily results in a mis-classification due to the lack of label information. Note that the label information used in the discriminator is only for better guiding the discriminator to distinguish distributions more accurately, and is not used for generating adversarial perturbation. As Figure 2 shows in the submission, the adversarial examples are generated according to the gradient of $L_d^0$ with respect to $x$, for which no label information is required.
>
>     (d) Thank you for pointing out this. In the revised version, we clarified these points. The manifold label is a binary value which indicates whether the latent features are induced by the perturbed samples or the clean samples. It is shown as $L_d$ in Eq.(6) and $L^0_d$ in Eq.(7). We also added the explanation for it in Figure 2.
>
>     (e) Roughly speaking, data manifold can be seen as the shape of the underlying distribution of the input data that lives in the input space. The input data are sampled from such manifold, where the inter-sample relationship may not be maintained when each data is considered separately. In the conventional adversarial training, each adversarial example is separately generated from a single clean data sample. Such  relationship might be implicitly captured by the neural network if the number of clean data samples tends to infinity; however it is impossible in practice. Given a finite number of training data, explicit consideration of inter-sample relationship helps, as demonstrated in this paper.

---

### Official Review · AnonReviewer2 · 2020-10-29
**Use global latent distribution to improve model robustness**

**Rating:** 7
**Confidence:** 2

**Review:**

The paper proposes a new method of improving model robustness by generating adversarial samples that are regularized by their latent distribution through f-divergence, whereas existing literature only uses local manifold property such as smoothness.

The method is well-motivated and the clarity of the paper is good. The experimental results are compared with several competitive baselines and the improvement looks significant (Although I am not familiar with the state-of-the-art experimental results).

Proofread is needed for the sentence "The adversarial examples are crafted by ... " on page 2 and several other small typos.

---

> ### Author Response · Authors · 2020-11-25
> **Response to AnonReviewer2**
>
> We highly appreciate the positive comments of the reviewer. We have revised the whole
> paper and corrected the typos and inappropriate expressions. Moreover, we have conducted two
> more experiments to show that our proposed method could achieve much better robustness than the
> latest competitive models against more updated attacks such as AutoAttack and Rays as shown in
> Appendix B.2

---

### Official Review · AnonReviewer4 · 2020-10-30
**Official Blind Review # 4**

**Rating:** 5
**Confidence:** 3

**Review:**

Summary: This paper considers the local and global information in adversarial attacks for adversarial training, where the authors design an adversarial framework containing a discriminator and a classifier. The idea is interesting and the paper is easy to follow.

However, I have still some concerns below:
- The novelty of this work combines the idea of PGD (local information) and Feature-Scatter (global information) .
- More importantly, the evaluation is no enough, even though Feature-Scatter considers the global information, but many attack methods have shown the robustness of Feature-Scatter was overestimated, such as [1][2][3] and so on. So I think evaluating on PGD and CW  is not enough.
- There are few analysis experiments for the proposed method, more analysis experiments are needed besides the comparision.

[1] Feature Attack: https://openreview.net/forum?id=Syejj0NYvr&noteId=rkeBhuBMjS

[2] RayS: A Ray Searching Method for Hard-label Adversarial Attack. KDD 2020.

[3] Reliable evaluation of adversarial robustness with an ensemble of diverse parameter-free attacks. ICML 2020.

---

> ### Author Response · Authors · 2020-11-25
> **Response to AnonReviewer4**
>
> Thank you for your efforts reviewing our paper and providing constructive comments
> and helpful suggestions.
>
>
> 1. Response to Cons1: Sorry for the confusion, but we respectfully disagree with this.
> To avoid possible confusion, we have revised the description of our method in the revised
> paper. Although Feature-scattering (FS) is one of our motivations to consider global information,
> our proposed ATDL method is fundamentally different from FS as well as the
> traditional AT. Specifically, FS generates adversarial examples by computing the feature
> matching distance between the batch of the original and perturbed samples, whilst our
> method leverages a discriminator to distinguish the whole latent manifolds resulted from
> the original clean and perturbed samples. In other words, our ‘manifold’ adversarial examples
> are crafted to disturb the manifold of latent distributions induced by the original
> samples as much as possible by leveraging the discriminator. We have revised the equation
> and added more descriptions to elaborate our proposed method more clearly.
>
>     It is worth noting that our method can be viewed as the game among three players: the
> classifier, discriminator, and adversarial examples (as emphasized in the revised version).
> The discriminator is learned to differentiate the latent distributions of the perturbed examples
> and clean data; the classifier is trained to (1) enforce the invariance between these two
> distributions to confuse the discriminator, and (2) classify the adversarial examples as accurately
> as possible; adversarial examples are crafted to differentiate the adversarial latent
> distribution from the natural one.
>
>
> 2. Response to Cons2: Following the suggestions, we have conducted additional experiments
> against AutoAttack and RayS on CIFAR-10 and CIFAR-100, which show our proposed
> ATDL-IMT+ can again outperform the existing state-of-the-art methods (as seen in Appendix
> B.2).
> Specifically, on CIFAR-10, our proposed ATDL-IMT+ method outperforms the-state-of-art
> methods by a large accuracy margin. Without exploiting additional data, our method can
> even perform better than all the other 9 algorithms: our method attains 70.60% accuracy
> against AutoAttack ($\epsilon = 8/255$) and 81.68% accuracy against Rays ($\epsilon = 8/255$), while the
> best of the others are just 65.88% and 64.6% respectively.
> On CIFAR-100, our ATDL-IMT+ could achieve 32.36% accuracy against AA ($\epsilon = 8/255$),
> which also outperforms all the other competitive methods without using additional data
> (e.g. without exploiting unlabeled data and pretraining). Although Gowal et al. (2020)*
> achieves better performance of 36.88%, it requires more unlabeled data. Moreover, our
> method is still ahead of its normal version which leverages no additional data.
>
>
> 3. Response to Cons3: There might be some misunderstandings about this. Due to the page
> limitation, several illustrative experiments and analysis are reported in Appendix including 1) how our proposed method affects the decision boundary compared with PGD and FS, 2)
> further analysis about the proposed ATDL-IMT, and 3) the illustration of the vector field
> for different perturbation schemes. Nonetheless, we will surely conduct more analysis in
> the final version.

---

> > ### Comment · AnonReviewer4 · 2020-11-25
> > **Thanks for the results.**
> >
> > Thanks for the additional results.
> > However, for me, the results are  still a bit strange and unconvincing.
> >
> > As shown in Table 1 and Table 3, for ATLD-IMT, the white-box attacks (acc: 74.46% ) are weaker than the black-box attacks (acc: 65.07%), which is exhibiting obfuscated gradients[1]. Could the authors show the evaluation results of ATLD-IMT under the Auto-Attack and the evaluation results of ATLD-IMT+ on the transfer-based attacks?
> >
> > In addition, could the authors evaluate the proposed methods with the adaptive attack, i.e., the proposed attack used in the training process? And how about the robustness under the  Feature Attack[2]?
> >
> >
> >
> > [1]  Obfuscated gradients give a false sense of security: Circumventing defenses to adversarial examples. ICML 2018.
> >
> > [2] Feature Attack: https://openreview.net/forum?id=Syejj0NYvr&noteId=rkeBhuBMjS

---

> > > ### Author Response · Authors · 2020-11-25
> > > **Response to AnonReviewer4**
> > >
> > > 1. Thanks for your further comments.
> > >
> > >   It is observed from Table 1 and Table 3 that this phenomenon can also be observed from the results of FS, i.e. the white-box attacks appear weaker than the black-box attacks. Note that the results of FS were obtained directly by running the source codes implemented by the authors of FS. Our source codes were also uploaded as complementary materials, which can be used for further verification.
> > >
> > >   We think the reason might be due to the unsupervised nature of FS and ATLD. Specifically, adversarial examples of both FS and our method are generated in an unsupervised fashion which can retain more structure information of data distribution. Therefore, capturing the manifold of adversarial examples, they could generalize better than the traditional AT methods on the white box attacks (as seen in Table 1). In the black-box setting (Table 3), the distributions of transferred adversarial examples may be inconsistent with the retained manifold information learned during the training of FS and ATLD. This may lead to an accuracy drop especially when compared with the white-box setting. Nonetheless, they still performed better than the other AT methods that do not consider the data manifold information.
> > >
> > >   We will leave this interesting topic for future investigations.
> > >
> > >
> > > 2. Following your suggestions, we have reported the ATLD-IMT against AA and Rays in the revised paper. Again, the ATLD -IMT is still ahead of many the-state-of-art methods (which can be seen in the updated Appendix).
> > >
> > >   It is noted that the results of ATLD -IMT in Table3 and Table6 of the original paper should be ATLD-IMT+ (which we have corrected in the updated version). While we feel sorry for just spotting such mistake, it actually won’t affect the conclusion at all since they performed more or less comparable in the black-box setting. It should also be noted that we discussed even in the original paper that the IMT method has a negative impact on transfer-based black-box attacks no matter ATLD-IMT or ATLD-IMT+.
> > >
> > >
> > > 3. Regarding the adaptive attack and feature attack as suggested, we don’t mind to conduct further experiments. However, the very limited rebuttal period may not allow us to do so especially since the suggestion was just made before the rebuttal window will be closed within a few hours. We will consider to add these comparisons in the final version.

---

### Official Review · AnonReviewer3 · 2020-10-31
**The paper analyzes the property of local and global data manifold for adversarial training.**

**Rating:** 7
**Confidence:** 4

**Review:**

The paper analyzes the property of local and global data manifold for adversarial training. In particular, they used a discriminator-classifier model, where the discriminator tries to differentiate between the natural and adversarial space, and the classifier aims to classify between them while maintaining the constraints between local and global distributions. The authors implemented the proposed method on several datasets and achieved good performance. They also compared with several whitebox and blackbox methods and proved superiority.

This paper was, in general, well written. The authors provided a good visualization of their analysis. Using local and global information for adversarial training is intuitive. The authors provided a good theoretical background to establish their method. The empirical evaluations show promising results.

Some major concerns are listed as follows:
1. It is not clear how equations 4 and 5 are realized using discriminator and classifier.
2. What kind of perturbations are chosen? It looks like all the experiments are with L-infinity. Does this observation hold for other ones?
3.  If the attackers leverage the global and local data manifold, can they bypass this attack?

---

> ### Author Response · Authors · 2020-11-25
> **Response to AnonReviewer3**
>
> We appreciate the positive comments of the reviewer and the constructive feedback.
> 1. Response to Cons1:
> Thanks for raising this issue. To clarify how Eqs.(4)-(5) are realized using discriminator and classifier, we have split Eq. (6) to two equations Eqs. (6)-(7) in the revised version. Note that now Eq. (6) is directly translated from Eqs. (4)-(5), where a discriminator network is employed for optimization, and it can be further reformulated in Eq. (7) by adding a regularization term to avoid some issues of discriminator networks. Specifically, in Eq. (6), we approximate Jensen-Shannon divergence between $P_\theta^*$($P_\theta$) and $Q_\theta$ with $\sup_W\sum_{i=1}^NL_d$, and minimize the classification loss on adversarial examples by minimizing$\sum_{i=1}^NL_f$ . As mentioned in the revised paper, it is a challenging task to evaluate the divergence between two latent distributions. To make it more tractable, we leverage a discriminator network for estimating the Jensen-Shannon divergence between two distributions $P_\theta^*/P_\theta$ and $Q_\theta$ according to Section 3.2.
>
>
> 2. Response to Cons2:
> Thanks for pointing out this issue. We use $L_\infty$ perturbation in all experiments including training and testing. We have added more description in the revised version.
>
>
> 3. Response to Cons3: The attacks which are unsupervised generated by our method may be
> weaker than the supervised ones such as PGD and CW, since our attacks did not leverage the
> label information and could not affect on gradients directly. Besides, since other methods
> have no components which aim to estimate the data manifold, so we could not generate
> global data manifold attacks on other methods which makes the results non-comparative.
>
>
> 4. Moreover, we have also conducted two more experiments to show that our proposed method could achieve much better robustness than the latest competitive models against more updated attacks such as AutoAttack and Rays as shown in Appendix B.2

---

### Author Response · Authors · 2020-11-25
**MAJOR REVISION**

We appreciate the constructive comments from the reviewers which indeed improve the quality of
our paper. In the revision, we substantially revised the paper, i.e., we conducted additional experimental
comparisons with several state-of-art methods against AutoAttack and Rays, made further
clarification to explain our technical contributions, proofread the paper, and corrected some inaccurate
expressions according to the comments from all the reviewers. For convenience, we have made
in blue all the revised parts in the new version.
1. Conducted more experiments. Overall, we have compared our ATLD-IMT+ with 9 other
competitive approaches against AutoAttack (AA) and Rays on CIFAR-10. Also, We have
compared ours with 4 competitive methods against AA on CIFAR-100. Experiments have
shown that our model significantly outperforms the other competitive models in these new
experiments (see Appendix for details).

    (a) Compared the proposed method ATLD-IMT+ against AA and RayS additionally with
9 other methods, i.e., WAR and RTS on CIFAR-10.

    (b) Compared the proposed method ATLD-IMT+ against AA additionally with four other
methods, i.e., Robust-overfitting and Pretraining on CIFAR-100.


2. Revised thoroughly to explain our technical contributions in equation (4), (5) and (6).


3. Proofreaded the paper and corrected typos and inappropriate expressions.

---

### Decision · Program_Chairs · 2021-01-07
**Final Decision**

**Decision:**

Reject

**Comment:**

This paper presents a framework for adversarial robustness by incorporating local and global structures of the data manifold. In particular, the authors use a discriminator-classifier model, where the discriminator tries to differentiate between the original and adversarial spaces and the classifier aims to classify between them. The authors implement the proposed approach on several datasets and the experimental results demonstrate performance improvements. The idea of using the global data manifold into addressing robustness of the learning model is interesting. However, the technical contribution and novelty have not been explained very well.